

# Effects of preferential flow on snowmelt partitioning and groundwater recharge in frozen soils

Aaron A. Mohammed[1], Igor Pavlovskii[1,2], Edwin E. Cey[1], and Masaki Hayashi[1]

[1]Department of Geoscience, University of Calgary, Alberta, T2N 1N4, Canada
5  [2]Golder Associates Ltd., Calgary, Alberta, T2A 7W5, Canada

*Correspondence to*: Aaron Mohammed (amohamme@ucalgary.ca)

**Abstract.** Snowmelt is a major source of groundwater recharge in cold regions. Throughout many landscapes snowmelt occurs when ground is still frozen, thus frozen soil processes play an important role in snowmelt routing, and, by extension, on the timing and magnitude of recharge. This study investigated the vadose zone dynamics governing snowmelt infiltration and groundwater recharge at three grassland sites in the Canadian Prairies over the winter and spring of 2017. The region is characterised by numerous topographic depressions where ponding of snowmelt runoff results in focused infiltration and recharge. Water balance estimates showed infiltration was the dominant sink (35–85%) of snowmelt under uplands (i.e. areas outside depressions), even when ground was frozen, with soil moisture responses indicating flow through the frozen layer. Refreezing of infiltrated meltwater during winter melt events enhanced runoff generation in subsequent melt events. At one site, time lags of up to 3 days between snowcover depletion on uplands and ponding in depressions demonstrated the role of shallow subsurface flow through frozen soil in routing snowmelt to depressions. At all sites, depression-focused infiltration and recharge began before ground thaw and a significant portion (45–100%) occurred while the ground was partially frozen. Relatively rapid infiltration rates and non-sequential soil moisture and groundwater responses, observed prior to ground thaw, indicated preferential flow through frozen soils. The preferential flow dynamics are attributed to macropore networks within the grassland soils, which allow infiltrated meltwater to bypass portions of the frozen soil matrix and facilitate both lateral transport of meltwater between topographic positions and groundwater recharge through frozen ground. Both of these flowpaths may facilitate preferential mass transport to groundwater.

## 1.0 Introduction

Snowmelt infiltration and soil freeze-thaw processes are important for modulating hydrological behavior in cold regions. In many northern landscapes the ground is still frozen when snowmelt begins, and consequently water movement into, through, and over frozen soil plays a critical role in routing snowmelt through catchments (Gray et al., 2001; Hayashi, 2013). Water movement in frozen soils is strongly affected by meteorological and soil moisture dynamics during the preceding seasons, and by the coupling of soil water flow and heat transfer processes (Stähli et al., 1999). This coupling, in addition to water




and energy transfer at the ground surface, creates complex subsurface flow dynamics during snowmelt (Ireson et al., 2013; Lundberg et al. 2016).

In many cold regions, focused infiltration of snowmelt under topographic depressions is an important mechanism of groundwater recharge (Baker and Spaans, 1997; Hayashi et al. 2003; French and Binley, 2004; Gerke et al 2010; Greenwood and Buttle, 2018). The Canadian Prairies located in the Northern Great Plains (Fig.1a) is an example of such an environment. The area has a cold semi-arid climate with snowmelt providing the major fraction of groundwater recharge (Maulé et al., 1994; Pavlovskii et al., 2018). The region is blanketed by glacial sediments with characteristic undulating terrain and many internally drained depressions (Winter and Rosenberry, 1998). During snowmelt, frozen ground limits infiltration under the uplands surrounding depressions, promoting runoff generation and ponding that results in depression focused infiltration and recharge (Hayashi et al., 1998). These depressions are an important component of subsurface water budgets as they control the snowmelt input to the groundwater system, and many depressions over the landscape collectively provide a large potential for recharge (van der Kamp and Hayashi, 1998; Berthold et al., 2004). Despite this significance, there are still uncertainties in the groundwater recharge function of prairie landscapes, specifically, in the combined effects of preferential flow and freeze-thaw processes on the rate and timing of meltwater infiltration and its partitioning between soil moisture and groundwater recharge prior to ground thaw.

Previous studies in cultivated fields in the Canadian Prairies have noted that the infiltration of ponded water in depressions begins when the soil is still partially frozen, but the initial infiltration rate is limited to the rate of soil thawing and groundwater recharge only begins once the soil completely thaws (Hayashi et al., 2003). However, perennial grasslands generally have a higher frozen soil infiltrability due to the presence and development of a macropore network, in comparison to croplands, where annual cultivation breaks up the macropore network near surface (van der Kamp et al., 2003). Macropores facilitate preferential flow that can cause water, solutes and thermal energy to bypass much of the soil matrix (Flury et al., 1994). Granger et al. (1984) classified one of the modes of infiltration into frozen soils as 'unlimited', where most available meltwater is infiltrated due to preferential flow through macropores. In unsaturated soils, macropores remain air-filled upon freezing, which can have a strong influence on the spatial and temporal characteristics of infiltration by enabling the transfer of water into and below the frost zone (Espeby, 1992; Ishikawa et al., 2006). Under these conditions, most infiltration occurs through macropores as freezing temperatures and pore-ice greatly reduce the hydraulic conductivity of the soil matrix (Holten et al., 2018; Grant et al., 2019; Demand et al., 2019). However, depending on soil temperature, infiltrated meltwater may be cooled by the surrounding frozen soil and refreeze along preferential pathways, blocking the flow of water to greater depths and reducing soil infiltrability (Stähli et al., 1996; Watanabe and Kugisaki, 2017). The influence of these processes on snowmelt partitioning and groundwater recharge dynamics in prairie grasslands remains unclear (van der Kamp et al., 1999; van der Kamp and Hayashi, 2009). An improved understanding of the interacting effects



of preferential flow and soil freeze-thaw processes, acting from the ground surface through the entire vadose zone, is needed to better understand their influence on groundwater recharge and snowmelt routing at hillslope and watershed scales.

Isolating the effects of frozen soil on meltwater partitioning remains challenging, as snowmelt-driven processes are further modulated by other local-scale factors that affect a field's water and energy balance, such as variability in topographic and microtopographic relief, wind-driven snow redistribution, shading and land cover (Stähli, 2005; Hayashi, 2013). These overlapping influences obscure the specific effects of frozen soil on snowmelt partitioning, thereby necessitating multi-site observations to draw generalized conclusions about water movement in frozen soil. The primary objective of this study was to characterize the infiltration and soil freeze-thaw processes controlling snowmelt partitioning simultaneously at three instrumented study sites, and evaluate the potential for both lateral and vertical preferential flow in frozen soils. The secondary objective was to examine how these processes affect the timing, magnitude and dynamics of groundwater recharge in prairie grasslands.

## 2.0 Study Region

Three perennial grassland sites (designated Stauffer, Spyhill and Triple G), located near the western edge of the Canadian Prairies, part of the larger Northern Prairie region (Winter and Rosenberry, 1998), were included in this study (Fig. 1). Surficial geology of the region surrounding the study sites is characterized by glacial sediments of varying composition, and hummocky or undulating terrain with numerous depressions (Fenton et al., 2013). The semi-arid climate is typical of the Canadian Prairies: cold winters, low annual precipitation (~ 400–500 mm) and hot, dry summers. Maximum frost depth is approximately 1 m, with freezing typically starting in November and thaw occurring in April to early May (Hayashi and Farrow, 2014). Due to proximity of the Rocky Mountains, foehn winds (locally known as "chinooks") are a common occurrence in the study area and lead to regular midwinter snowmelt events when air temperatures rise above 0 ºC (Nkemdirim, 1996).

Studies focused on an individual drainage catchment and corresponding depression at each site (Figs. 1b–d): SE2 at Stauffer; W at Triple G; and GP at Spyhill. At the time of the study, land cover was grassland used for cattle grazing at Stauffer and Triple G, and ungrazed grassland at Spyhill. Soils at the sites are classified as Orthic Black or Brown Chernozems (Soil Classification Working Group, 1998) underlain by glacial till of varying composition ranging in thickness from 7–20 m (Pavlovskii et al., 2018). The present study focuses on monitoring of surface and subsurface hydrological conditions along a depression-upland complex at each site using data collected during the winter and spring of 2016–2017.



## 3.0 Methods

### 3.1 Meteorological and land surface measurements

Each site was instrumented with a meteorological station equipped with sensors measuring air temperature, humidity, wind speed, net radiation, total (rain and snow) precipitation, and an eddy-covariance system consisting of a sonic anemometer and a krypton hygrometer at 1.9 m above ground surface. All-season precipitation data for Triple G site was not available, so precipitation records from an Alberta Agriculture and Forestry weather station located 5 km north (Standard AGCM) was used instead. Details regarding instrument models and precipitation and eddy covariance data processing are described in Pavlovskii et al. (2019b). At Spyhill and Triple G, the meteorological station was located directly above the upland instrumented soil pit within the internal catchment of depressions GP and W, respectively while at Stauffer it was situated approx. 600 meters north of depression SE2. Snowcover and water ponding conditions were monitored using time-lapse cameras (Wingscapes, TimelapseCam) and snow gauges at upland and depression locations. To monitor the ponding and recession rates in depressions, water levels were recorded on snow gauges visible in the camera images. Water depths in depressions were also measured using pressure transducers (Solinst Levelogger 3001) when possible. In addition, snow surveys were conducted at various intervals throughout the winter. Snow depth was measured using a metal ruler along a 100 to 200 m snow course traversing each depression and surrounding uplands, and snow samples were collected at 50-m intervals using a 7.0-cm diameter aluminum snow tube. Snow samples were transferred to a sealable bag and weighed in the laboratory to determine snow water equivalent (SWE). The average SWE was calculated from the average snow depth and density.

Snowmelt runoff was estimated from the volume of water collected in depressions (Mohammed et al., 2013). Pond water levels in depressions were converted to runoff volumes using depth-volume relationships derived from high-resolution topographical surveys of the depressions. The effective snowmelt runoff per unit area was then calculated by dividing the volume of water by the catchment area. This method underestimates runoff as it does not consider water that initially infiltrates within the depression prior to observable ponding (Hayashi et al., 2003).

### 3.2 Vadose zone and groundwater monitoring

Vadose zone conditions were monitored using instrumented soil pits recording soil moisture and temperature at upland and depression locations. Details of the depth and intervals of subsurface instrumentation are listed in Table 1. Required excavations and instrument installation were carried out in October 2014, so disturbance of the soil profile is expected to be minimal for the 2016-2017 season reported here.

Time-domain reflectometry (TDR) probes (connected with SDMX multiplexers, Campbell Scientific) were used to estimate unfrozen soil moisture under uplands and depressions at Stauffer and Spyhill sites and in the depression at Triple G. Similar



to Seyfried and Murdock (1996), the empirical equation of Topp et al. (1980) was used to calculate liquid soil moisture from the apparent bulk dielectric constant and laboratory testing confirmed the accuracy of this approach in comparison to more physically-based dielectric mixing models (LeBlanc, 2017) . The upland soil pit at Triple G utilized capacitance probes (Stevens, HydraProbe II) to measure liquid soil moisture, which were individually calibrated following the procedure

outlined by Hayashi et al. (2010) using collocated TDR probes that were manually measured with a portable TDR unit (Soil Moisture Equipment, Trase). Soil temperature at corresponding depths were measured by Type T thermocouples. All vadose zone data were recorded at hourly or half-hourly intervals using dataloggers (Campbell Scientific, CR1000, 10X, and 23X). A number of thermocouples and TDR probes were permanently damaged and were excluded from any analysis (Table 1). Additionally, several TDR probes malfunctioned during winter and early spring at SE2 depression at Stauffer with

corresponding data excluded from the analysis. Soil core samples (5cm diameter by 5cm length) from Stauffer underwent laboratory testing using falling head permeameters to determine saturated hydraulic conductivity ($K_{sat}$).

Vertically nested piezometers, installed in 2014, were used to monitor groundwater levels beneath depression and uplands. Only results from the shallowest piezometers are shown in this study. Details of screen lengths and depths are included in

Table 1. Water levels in piezometers were recorded by pressure transducers every 30 minutes (Solinst Levelogger 3001; InSitu miniTROLL). Additionally, manual water level measurements were performed on irregular intervals to calibrate transducers and ensure no instrument drift. $K_{sat}$ was calculated from slug tests (Hvorslev, 1951) performed on the piezometers.

**3.3 Snowmelt-event water balances**

Water balance calculations were used to assess the significance of the various water flow pathways and partitioning of snowmelt at the ground surface over individual snowmelt events. In order to close the water balance over each melt event, only snowmelt events in which complete depletion of the snowpack occurred were utilized. This allowed the amount of infiltrated water to be estimated as the residual of the water balance:

$$I = \Delta SWE + P - R_o - E_v \tag{1}$$

where $I$ is infiltration, $\Delta SWE$ is change in snow water equivalent over the melt event as estimated from snow surveys, $P$ is precipitation from the date of the last snow survey to the end of the snowmelt event, $R_o$ is runoff estimated from the change in ponded volume within depressions, and $E_v$ is vapour flux (i.e. sublimation, evapotranspiration, an/or condensation) measured by the eddy-covariance system. All components are reported in millimetres (mm).

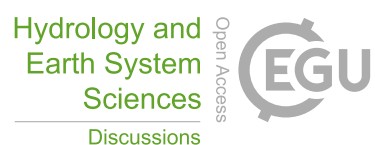

## 4.0 Results

Air temperature dynamics were similar at all study sites, showing consistent midwinter periods when air temperatures rose above 0 ℃. We will focus on individual warming events when snowmelt occurred: midwinter warming events (referred to as "MW") and spring snowmelt (referred to as "Spring"). Soil freeze-thaw conditions at the study sites are classified in terms of

three soil temperature states, designated as frozen (soil temperature at all depths within the frost zone is below -0.5 ℃), partially frozen (at least one soil temperature sensor within the frost zone is between -0.5 and 0.5 ℃) and thawed (all soil temperature probes within the frost zone above 0.5 ℃). The frost zone here is defined as a seasonally-frozen part of the soil profile and extends from surface to approximately 1 m depth.

### 4.1 Stauffer Site

The first midwinter snowmelt event (MW1) occurred on January 17 (Fig. 2a). At the upland position, soil moisture and temperature promptly responded to the warming with signals propagating to a depth of 1 m (Fig. 2b and c), suggesting some infiltrated water reached soil below the frost zone. The event is followed by simultaneous decreases in soil temperatures and liquid moisture above 1 m depth, consistent with the refreezing of infiltrated water in the frost zone. Snowcover depletion was not complete during MW1 at this site, so the water balance could not be closed, and no infiltration ratio was calculated.

At the same time, absence of ponding and limited soil moisture response in the depression (Fig. 3a and d) points to a runoff ratio of zero.

Melt event MW2 occurred on January 31 but was very brief and resulted in only minor thermal and moisture responses under both landscape positions.

The MW3 event occurred on February 16 and caused complete snowcover depletion on uplands and ponding in the depression (Fig. 3b). Water balance components for this event are shown in Table 2. The freezing of the pond surface prevented observations of the pond recession. However, an air-filled space was observed beneath the ice on March 3, indicating that all unfrozen water had infiltrated. Depression ponding during MW3 was associated with sharp increases in

liquid soil moisture at depths up to 1.5 m (i.e., below the frost zone) followed by a rise in the groundwater level (Fig. 3d and e). These soil moisture responses below the frost zone occurred just before temperatures in the frozen zone reached 0 ℃ (Fig. 3c), indicating that pore-ice was still present in the frost zone when recharge occurred. After the MW3 event, the decrease in liquid moisture at depths of up to 0.3 m with soil temperatures remaining at 0 ℃ (Fig. 3c) is consistent with refreezing of previously infiltrated water.

The Spring snowmelt event occurred on March 21 and again resulted in complete snowcover depletion and runoff ponding in the depression (Fig. 3b). Water balance calculations for this event showed an increased runoff ratio compared to MW3





(Table 2). Concurrent with ponding (March 21), soil moisture increased below the frost zone (1.0 and 1.5 m) along with a rise in groundwater levels, indicating that meltwater was infiltrating even though pond levels remained stable (Figs. 3d and e). Non-sequential wetting was also observed as soil moisture and groundwater level increased below the frost zone despite little to no soil moisture response at 0.1 to 0.5 m depths (Fig. 3d), which were still partially frozen (Fig. 3c). The pond water

level recession began on March 26 at a rate of 50 mm d$^{-1}$ ($6 \times 10^{-7}$ m s$^{-1}$). The geometric mean of $K_{sat}$ for soil samples at depths of 1.0–2.5 m measured with core permeameters was $1 \times 10^{-7}$ m s$^{-1}$. Considering core permeameter tests tend to underestimate field-scale hydraulic conductivity, the pond recession rate is comparable to the $K_{sat}$ of the subsoil beneath the frost zone under the assumption that ponded infiltration takes place under a unit hydraulic gradient. The ground was still partially frozen by the time all ponded water infiltrated and remained so until complete ground thaw on April 20, 30 days

after the Spring recharge event began (Fig. 3c).

The ground beneath the upland remained frozen during all MW events (Fig. 2b), with infiltrated water refreezing between events (Fig. 2c) and did not completely thaw until April 22. Time-lapse cameras showed that there was complete snowcover depletion by March 24, indicating that all snowmelt occurred while the ground was still frozen, and the frost thickness was

approximately 1 m (Fig. 2b). During the Spring event, all upland soil moisture sensors showed increases in soil moisture relative to pre-freezing conditions. A non-sequential response was observed as the 1 m sensor showed an increase prior to shallow sensors (Fig. 2c). There was a slight increase in hydraulic head beneath the upland (Fig. 2d), but this rise is interpreted as a response due to groundwater mound formation under the depression causing lateral flow to adjacent uplands.

### 4.2 Triple G Site

Runoff and ponding in depression W occurred during all four warming events (Fig. 4b). Unlike Stauffer, visual inspection showed that the ponded water in the depression completely froze after each ponding event and the pond water level did not decline between events (Fig. 4b). Complete snowpack depletion on the uplands during MW1, MW3, and Spring events allowed the water balance to be closed. Calculations show that each subsequent melt event had an increase in runoff ratio. (Table 2).

Liquid soil moisture at 0.2 and 0.8 m under the depression gradually increased to above pre-freezing values between events MW1 and MW2 (Fig. 4d) along with a slight response in groundwater level (Fig. 4e), indicating some infiltration had taken place. After MW2, liquid soil moisture at 0.2 m decreased simultaneously with a temperature decrease to below 0 °C, consistent with refreezing of the infiltrated water. No event-specific changes in liquid soil moisture and soil temperature

occurred during MW3 (Fig. 4d) despite the rise in the pond level (Fig. 4b), suggesting no infiltration in the depression occurred during this event due to refreezing of meltwater at 0.2 m after MW2.





The ponded water level recession started on March 22 at an average rate of 15 mm d$^{-1}$ (2.0×10$^{-7}$ m s$^{-1}$) (Fig. 4b). Groundwater recharge occurred on March 24 (Fig. 4e) and coincided with the thawing of the top 0.2 m of the soil while the soil was partially frozen between 0.4 – 1.0 m depth (Fig. 4c). The pond recession rate did not change once the soil thawed completely on March 28 (i.e. the infiltration rate during partially frozen ground conditions was the same as the infiltration rate during unfrozen conditions) with approximately 45% of the water in the pond infiltrated before the soil had completely thawed. After April 9, when the water level in the piezometer reached the ground surface of the depression, the pond recession rate slowed to 6.5 mm d$^{-1}$ (7.5×10$^{-8}$ m s$^{-1}$) (data not shown). From April 3, the hydraulic head under depression W remained stable during pond recession (Fig. 4e). The observed recession rates were up to an order of magnitude larger than the hydraulic conductivity of 2.0×10$^{-8}$ m s$^{-1}$ measured from slug tests on the sediments at approximately 6 m depth, however hydraulic conductivity of shallower soil within and directly below the frost zone is expected to be higher due to increased weathering (Hayashi et al., 1998).

On several occasions, time delays were observed between snowcover depletion on the uplands and the corresponding pond level rise in the depression (Fig. 5). For example, during MW1, most snowmelt occurred on January 17 and only lower parts of the slopes retained snow by the morning of January 18. Pond levels increased throughout January 18 despite very limited change to the remaining snow-covered area. The largest time delay occurred during spring snowmelt, when most snowcover disappeared on March 14 and 15, while the pond level in the depression rose by ~ 0.1 m on March 18 (Figs. 5c and S3). The other observed time delays were shorter (Fig. 5a and b; and Figs. S1 – S2).

The upland soil profile remained at least partially frozen during all melt events, with soil thaw beginning on March 20 and becoming completely thawed on April 13 (Fig. 6b). All upland soil moisture sensors up to 1m depth showed increased soil moisture relative to pre-freezing conditions, however almost no change at 1.5 m indicated a lack of deep flow over the MW and Spring events. During the entire winter and spring snowmelt period, no groundwater response was observed under the upland (Fig. 6d).

## 4.3 Spyhill Site

No ponding of runoff was observed in depression GP during midwinter snowmelt events. At the same time, complete snowpack depletion on the uplands during MW1, MW3, and Spring events allowed calculation of the water balance (Table 2). Despite infiltration of over 80% of snowmelt derived from the water balance during both MW events, soil moisture probes only showed responses to event MW3. During this event, the liquid soil moisture increased under the depression at depths up to 0.5 m (Fig. 7d) and upland throughout the entire profile (Fig. 8c). Soil temperatures became negative at depths up to 0.5 m after event MW3 under both the upland and depression (Fig. 7c, 8b) with corresponding drops in liquid soil moisture consistent with refreezing of infiltrated water (Fig. 7d, 8c).



The Spring snowmelt event occurred on March 15 and caused ponding within depression GP (Fig. 7b). The ponding coincided with an increase in liquid soil moisture at 0.1 m depth (Fig. 7d) and onset of groundwater recharge (Fig. 7e). The liquid soil moisture at 0.3 and 0.5 m remained relatively stable until after complete pond infiltration. Recession of the pond level accelerated on March 19, reaching a maximum rate of circa 34 mm $d^{-1}$ ($4\times10^{-7}$ m $s^{-1}$), which approximated the

hydraulic conductivity of $6\times10^{-7}$ m $s^{-1}$ at 3 m depth measured by slug tests. The pond in the depression disappeared on March 22, whereas complete ground thaw did not occur until a month later on April 22, indicating all infiltration occurred before ground thaw.

The Spring event caused a simultaneous increase in liquid soil moisture throughout the upland profile (Fig. 8c), all sensors

showed an increase relative to pre-freezing conditions, indicating percolation to a depth of at least 1.5 m during the Spring event. Time-lapse cameras confirmed that snowcover was fully depleted by March 14, indicating all snowmelt occurred while the ground was still frozen. The groundwater level in the upland piezometer showed a change in upward trend coinciding with complete thaw of the ground (Fig. 8d), which may be indicative of recharge under uplands and/or of groundwater mound formation under the depression causing lateral flow to adjacent uplands. The reason for the upward

trend in hydraulic head prior to ground thaw is attributed to a recharge event from the previous year, highlighting the piezometer's extremely long response time (on the order of months) (Fig. 9).

## 5.0 Discussion

### 5.1 Snowmelt partitioning

The infiltrability of frozen soils strongly affects the partitioning of snowmelt into infiltration or runoff, and thus, quantity of

water available for groundwater recharge under depressions. Snowmelt event water balances (Table 2) show that the majority of snowmelt was infiltrated under uplands despite the ground being frozen/partially frozen. During all MW events, the ground within the frost zone underneath the uplands at all sites remained at or below 0 °C, indicating that even when the ground was frozen, infiltration was still the dominant sink of snowmelt fluxes. The snowmelt runoff ratios from all sites ranged between 0 and 57%, with a mean of 21%. These values are lower than those reported by Hayashi et al. (1998) for a

cultivated depression-upland catchment in the Canadian Prairies over the winter and spring periods of 1993 to 1996, who reported snowmelt runoff ratios ranging from 28 to 60% with an average of 43%. The lower runoff ratios in this grassland study are expected as the lack of cultivation would allow the formation of well-developed macropore networks and higher soil infiltrability (van der Kamp et al., 2003; Bodhinayake and Si, 2004). In a more humid climate, Greenwood and Buttle (2018) reported average snowmelt runoff ratios for contrasting grassland and cultivated depression-upland catchments of

52% and 72%, respectively. The authors also attributed the lower runoff in grassland catchments to enhanced macropore flow.



When snowmelt occurs over frozen soils, infiltration-runoff partitioning is strongly affected by the air-filled porosity and amount of unblocked preferential flow pathways in the upper tens of cm of soil at the time of snowmelt (Gray et al., 2001) as well as the temperature of the frozen soil (Stähli et al. 1999). The relatively dry soils at the beginning of winter are characterised by a relatively high infiltrability, as evident at all sites by the lower runoff ratios during MW1, and MW events having lower runoff ratios than Spring melt events (Table 2). The frozen soil early in the winter would have a relatively high air-filled porosity and low ice content, and thus would offer little obstruction to meltwater infiltration, as is typical for dry frozen soils (Kane and Stein, 1983; Granger et al., 1984; van der Kamp et al., 2003). Macropores in this situation would be air-filled and open for infiltration after freezing (Espeby, 1992). However, refreezing of infiltrated water in the soil matrix and macropores near the infiltration source (i.e., ground surface) reduces infiltrability and flow-path connectivity due to pore-blockage with ice (Stähli et al. 1996; Watanabe and Kugisaki, 2017). Consequently, a drop in soil temperature below 0 °C between MW events (Figs. 2b, 6b, 8b) and associated refreezing of infiltrated water would reduce infiltrability and, thus, increase runoff during subsequent melt events. Such an effect likely contributed to the sequential increase in runoff ratios during melt events at Stauffer and Triple G sites (Table 2). These observations of refreezing of infiltrated meltwater are generally consistent with the 'pre-fill' phase of the conceptual model of runoff generation on frozen soil proposed by Appels et al. (2018), in which the refreezing of initially infiltrated water is necessary to reduce infiltrability before runoff can occur. To the best of our knowledge, no other study has demonstrated the time sequencing of this subsurface infiltration/refreezing effect on enhancing subsequent runoff partitioning over multiple snowmelt events.

At Spyhill, no runoff was produced during MW events and >80% of snowmelt infiltrated under uplands over all events (Table 2). A possible explanation for the enhanced infiltration is that grasslands at Spyhill are ungrazed, in contrast to the other two sites. Grazing has been shown to compact the soil surface and reduce air-filled porosity and macroporosity in the upper-most 50 cm of the soil (Naeth et al., 1991). Grasses allowed to grow over multiple years also tend to have deeper rooting depths and macropore-networks compared to grazed fields (Naeth et al., 1990a; 1990b). Both these effects would increase the infiltrability and preferential flow paths within the frozen soil, resulting in higher infiltration ratios at Spyhill. The conversion of fields to ungrazed grasslands has also been shown to cause the drying out of prairie wetlands by drastically reducing snowmelt runoff due to increased infiltration capacity of frozen soils (van der Kamp et al., 1999).

Observed delays between snowpack depletion and depression ponding (Fig. 5) suggest that preferential pathways can play an important role in runoff routing on a hillslope scale. Very dry soils at the Triple G site would allow near-surface macropores to remain open and connected, allowing subsurface flow along the topographical gradient from upland to depression. Granger et al. (1984) mentioned this as probable mechanism for snowmelt runoff in frozen prairie soils but had no measurement to verify the proposition. Following snowmelt depletion after each event, the ground under the upland remained frozen and soil moisture probes below the frost zone showed no response (Fig. 6b and 6c). Groundwater levels showed no response (Fig. 4e and 6d), ruling out any groundwater upwelling to the depression surface. This indicates there



must be a mechanism in the frozen soil that simultaneously perches infiltrated meltwater while also providing a pathway to transport it downslope. Shallow subsurface flow can be a major runoff mechanism in structured soils, where the presence of a low permeability horizon underlying a more permeable layer causes a portion of vertically percolating water to be deflected laterally, and transported downslope (Weiler et al., 2006; Chifflard et al., 2019). Macropore-rich soil horizons

overlaying lower permeability soils or bedrock have been identified as an environment for this type of lateral subsurface flow (Whitson et al., 2004; Redding and Devito, 2010). At the Triple G site, the preferential pathways contributing to infiltration are probably interconnected at shallow depth (0–0.2 m) as this is the zone with the highest density of macropores (LeBlanc 2017). Macropores may provide a conductive subsurface pathway above a less conductive layer (in this case, frozen soil with less macropores). Thus, subsurface flow, in addition to overland flow, can be a factor in snowmelt routing

on prairie hillslopes.

In terms of groundwater recharge, given the lack of groundwater level responses under uplands at Stauffer and Triple G (Figs. 2d and 6d), upland contribution to the recharge is most likely negligible at these sites, consistent with other studies in the region (Hayashi et al., 1998; Berthold et al., 2004). However, high upland infiltration ratios and soil moisture responses

below the root zone at Spyhill (Table 2 and Fig. 8c) suggest that diffuse recharge may be possible under undisturbed prairie grasslands, as identified in chloride mass-balance estimates at the site (Pavlovskii et al., 2019a) and earlier studies that initially discussed diffuse recharge as a consequence of the conversion of agricultural fields to undisturbed grasslands (van der Kamp and Hayashi, 1998; 2009).

**5.2 Focused infiltration and preferential flow dynamics**

There were several observations linking infiltration within ponded depressions to preferential flow through frozen ground (Table 3). At all sites during the Spring event, infiltration and recharge began prior to complete ground thaw and occurred despite the presence of an approximately 1 m thick frost zone (Figs. 3, 4, 7). At Stauffer and Spyhill, 100% of ponded water infiltration took place while the soil was partially frozen, while at Triple G, 45% of the runoff collected in depression W infiltrated before complete ground thaw. This number likely represents more than 50% of the total infiltration amount as

evaporation rates increase with higher radiation and air temperature in spring and more pond water will be lost to the atmosphere and unavailable to infiltrate (Hayashi et al., 1998).

The evidence for preferential flowpaths includes the detection of non-sequential wetting (Allaire et al., 2009), indicated by the non-sequential increases in liquid soil moisture or groundwater recharge responses during focused infiltration events

(Table 3). These responses show that surface water bypassed the frost layer. Furthermore, this flow occurred when ponding began and near-surface (0.1–0.2 m) layers became saturated (Figs. 3d, 4d, 7d), consistent with other studies that concluded that near-saturated conditions in partially frozen topsoil during snowmelt can supply water to deeper macropores (e.g. Ishikawa et al., 2006; Scherler et al., 2010). Groundwater recharge events occurred when soil in the frost zone was still





partially frozen, days to weeks before complete ground thaw. The observations reported here suggest that the ponding of water and filling of the unfrozen pore-space can activate preferential flow pathways and cause channeling of snowmelt to groundwater before complete thawing of the soil profile.

Further evidence for a dominant role of preferential flow includes relative independence of the pond recession rates from soil frost condition. A clear illustration of this can be seen at Triple G, where the pond infiltration rate remained steady as the soil transitioned from partially frozen to unfrozen (Fig. 4b), even though soil freezing is normally associated with a decrease in hydraulic conductivity by orders of magnitude for non-macroporous soils (Burt and Williams, 1976; Watanabe et al., 2013). At two other sites (Stauffer and Spyhill) pond recession rates were similar to the $K_{sat}$ of the subsoil, despite the presence of a

layer of partially frozen soil layer 0.8 – 1.0 m thick. The infiltration rate in all cases was limited by the hydraulic conductivity of the layers beneath the frost zone, suggesting the existence of a mechanism facilitating rapid flow through the frost zone that is consistent with the presence of macropores. This finding contradicts a previous study in cropped fields (Hayashi et al., 2003) that noted the pond infiltration rate was limited to the soil-thaw rate when soil frost was present, and then increased to values similar to the subsoil $K_{sat}$ when unfrozen, giving rise to groundwater recharge only after complete

ground thaw. The likely explanation for this inconsistency is that tillage in cropped fields destroyed shallow macropore networks that enable preferential flow through the frost zone. Enhanced infiltration in frozen macroporous versus non-macroporous soils has been shown by van der Kamp et al. (2003). Observations here indicate that when a macropore network is present, in addition to enhanced infiltration at the ground surface, deeper percolation of meltwater and groundwater recharge can occur prior to ground thaw as well.

There was also evidence of refreezing of infiltrated meltwater in depressions following MW events, which limited over-winter recharge by preventing water migration deeper within the soil profile (Fig. 4d and 7d). When ponding occurs, infiltration and refreezing takes place in the largest (previously air-filled) pores. Whether refreezing occurs depends on the weather following snowmelt and the energy-input to the pond (Pavlovskii et al., 2019b). Thermal energy supplied by

increased solar radiation during spring is sufficient to prevent refreezing of water in the pond and subsurface, allowing macropores to transport water to the unfrozen soil below.

**5.3 Hydrological implications of preferential flow in frozen soils**

As shown above, preferential flow in frozen soils plays an important role in both vertical flow through the soil and lateral transport of meltwater between topographical positions. Observations indicate that rates of ponded water infiltration into

frozen soil are decoupled from the hydraulic conductivity of the frozen soil matrix and are largely dependent on the hydraulic conductivity of soil beneath the frost zone. Such decoupling means that infiltration rates can routinely surpass the typically low hydraulic conductivity values of the frozen soil matrix. This enables large volumes of ponded water to infiltrate deeper in the soil profile relatively early in the spring season, before transpiration begins and when evaporation is still low.



Thus, this initial recharge pulse through frozen ground may escape the effects of evapotranspiration and shallow groundwater cycling and contribute disproportionately to deeper recharge. Recent studies investigating the isotopic nature of groundwater in aquifers across the Canadian Prairies have noted the isotopic composition shows the prevalence of winter precipitation which has not been subject to significant evaporation (Stumpp and Hendry, 2012; Pavlovskii et al., 2018; Bam,

5 2018), even though ponded water in depressions (from snowmelt) will exhibit an evaporative signature later in spring (Pavlovskii et al., 2018). The preferential recharge of this early snowmelt infiltration shown here may be an important mechanism responsible for this phenomenon. Furthermore, by facilitating a reduction in ponded volume of depressions, preferential flow reduces the connectivity of ponds in depression-dominated catchments, thereby influencing 'fill and spill' runoff routing and streamflow generation on a watershed scale (Shaw et al., 2012; Brannen et al., 2015).

Similarly, snowmelt infiltration (and, by extension, runoff generation) in the uplands appears to be controlled by an interplay between snowmelt inputs, soil temperature, and preferential flow dynamics, with the latter being controlled by the antecedent soil moisture and infiltration-refreezing history of previous snowmelt events. Uplands that remain frozen during snowmelt events cause the refreezing of infiltrated water, which reduces soil infiltrability and allows greater runoff

generation to depressions during later events. In addition, the observed delays between snowmelt and depression ponding indicates that subsurface flow is an important runoff mechanism on frozen hillslopes. How this process affects the timing and magnitude of runoff generation compared to overland flow in these environments (e.g. Coles and McDonnell, 2018) is unclear and warrants further investigation.

From a water quality perspective, preferential flow creates a pathway from surface to groundwater that bypasses much of the soil matrix. Such bypass flow has the potential to facilitate transport of labile nutrients and surface applied chemicals to groundwater during snowmelt (e.g. Grant et al., 2018). In contrast to preferential flow in unfrozen soils, there are limited opportunities for mass exchange with the frozen matrix, which can further reduce the filtering effect of the soil (Holten et al., 2018).

**6.0 Conclusions**

This study highlighted the effects of preferential flow in frozen soils and infiltration-refreezing mechanisms on the hydrologic functioning and winter water balance of prairie grasslands. Despite the ground remaining frozen throughout snowmelt events, infiltration into frozen soil was the major sink of snowmelt at all three sites, and modulated the amount of runoff available for depression-focused recharge. In addition, focused infiltration and preferential flow in frozen soil enabled

meltwater to bypass a portion of the soil profile and groundwater recharge prior to ground thaw. Field data suggested that the refreezing of infiltrated meltwater during midwinter snowmelt contributed to runoff generation in frozen grassland soils, highlighting the feedback effect of previous melt events on later snowmelt partitioning. Time delays between snowcover

depletion and ponding in depressions demonstrated that shallow subsurface flow, in addition to overland flow, can be an important runoff mechanism on frozen prairie hillslopes. Both of these flowpaths may facilitate preferential mass transport to groundwater.

Midwinter snowmelt events are a regular occurrence in the Canadian Prairies and other seasonally frozen landscapes (Bayard et al., 2005; Sutinen et al., 2008; Pavlovskii et al., 2019b) and are projected to increase in frequency and magnitude due to climate change (Henry, 2008). Thus, a better understanding of the dynamics of subsurface processes in response to such meteorological forcing is required to predict the effects of melt events on water partitioning in cold regions. However, the coupled hydrodynamic mechanisms and thermal processes are still not clearly understood (Mohammed et al., 2018). Further

research investigating preferential flow and soil-freeze thaw interactions is needed to understand how these processes will affect the winter water balance and cold region hydrological processes under different land use and climate conditions.

**Acknowledgements**

The authors wish to thank Shelby Snow, Brandon Hill, and Polina Abdrakhimova for field assistance; Larry Bentley and Saskia Noorduijn for helpful discussions; and Alberta Innovates, Alberta Environment and Parks, Alberta Energy Regulator

- Alberta Geological Survey, and the Natural Sciences and Engineering Research Council of Canada for funding support.

*Author contributions*: AM installed vadose zone instrumentation and piezometers at study sites, collected data, processed and analysed data (except eddy-covariance data), and wrote the paper. IP performed snow surveys and processed eddy-covariance data, and provided recommendations and revisions to the paper prior to submission. EC oversaw data collection

and analysis, and provided numerous recommendations and revisions to the paper prior to submission. MH installed weather stations at the study sites and provided recommendations and revisions to the paper prior to submission.

*Competing interests*: The authors declare that they have no conflict of interest.

*Data availability*: The data used to calculate values in Table 2 and are provided as a table in the Supplement.

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



## Tables

**Table 1 - Subsurface instrument depths and piezometer screen intervals.**

| Study site | Location | Instrument | Depth (mbgs) |
|---|---|---|---|
| **Stauffer** | depression, upland | WC, TC | 0.1, 0.2[†], 0.3, 0.5, 1.0[‡], 1.5 |
| | depression | piezometer | 5.62/6.38[±] |
| | upland | piezometer | 11.25/12.75[±] |
| **Triple G** | depression, upland | WC, TC | 0.2, 0.4, 0.6, 0.8, 1.0[†], 1.5[‡¶] |
| | depression | piezometer | 2.62/3.38[±] |
| | depression | piezometer | 6.25/7.75[±] |
| | upland | piezometer | 15 14.25/15.75[±] |
| **Spyhill** | depression, upland | WC, TC | 0.1[†], 0.2[¶], 0.3, 0.5, 1.0[¶], 1.5 |
| | depression | piezometer | 2.62/3.38[±] |
| | upland | piezometer | 8.65/10.15[±] |

TC – thermocouple, WC – soil moisture probe, mbgs – meters below ground surface, † – upland TC damaged, ‡ – depression TC damaged, § – upland WC damaged, ¶ – depression WC damaged, ± – numbers in depth column for piezometers represent screen top and bottom depths, respectively.



**Table 2 - Water balance components for individual snowmelt events at study sites.**

| Site | Event | ΔSWE (mm) | P (mm) | R (mm) | E (mm) | I (mm) | IR % | RR % |
|------|-------|-----------|--------|--------|--------|--------|------|------|
| **Stauffer** | MW1 | -- | | 0 | | -- | -- | 0 |
| | MW3 | 37 | 3 | 12 | 2 | 25 | 64 | 31 |
| | Spring | 15 | 26 | 14 | 9 | 18 | 43 | 33 |
| **Triple G** | MW1 | 26 | 0 | 4 | 1 | 22 | 83 | 14 |
| | MW3 | 22 | 0 | 9 | 2 | 11 | 52 | 42 |
| | Spring | 0 | 14 | 8 | 1 | 5 | 35 | 57 |
| **Spyhill** | MW1 | 20 | 7 | 0 | 5 | 22 | 82 | 0 |
| | MW3 | 25 | 0 | 0 | 4 | 21 | 85 | 0 |
| | Spring | 10 | 3 | 1 | 2 | 10 | 83 | 8 |

IR – infiltration ratio; RR – runoff ratio.



**Table 3 - Observations of preferential flow in frozen soil under depressions at study sites.**

| Warming event | Study site | | |
|---|---|---|---|
| | Stauffer (SE2) | Triple G (W) | Spyhill (GP) |
| **MW3** | Increases in liquid soil moisture at 1 and 1.5 m (under frost zone) | | Increases in liquid soil moisture both within and under frost zone |
| **Spring** | Increases in liquid soil moisture at 1 and 1.5 m (under frost zone) before similar increases in the zone directly above<br><br>Complete ponded water infiltration prior to soil thaw<br><br>Groundwater level increase prior to soil thaw | Infiltration rate in partially frozen soil equals infiltration rate in unfrozen soil<br><br>Groundwater level increase prior to soil thaw | Increase in groundwater level before soil moisture increases at 0.3 and 0.5 m<br><br>Complete ponded water infiltration prior to soil thaw<br><br>Groundwater level increase prior to soil thaw |

**Figures**

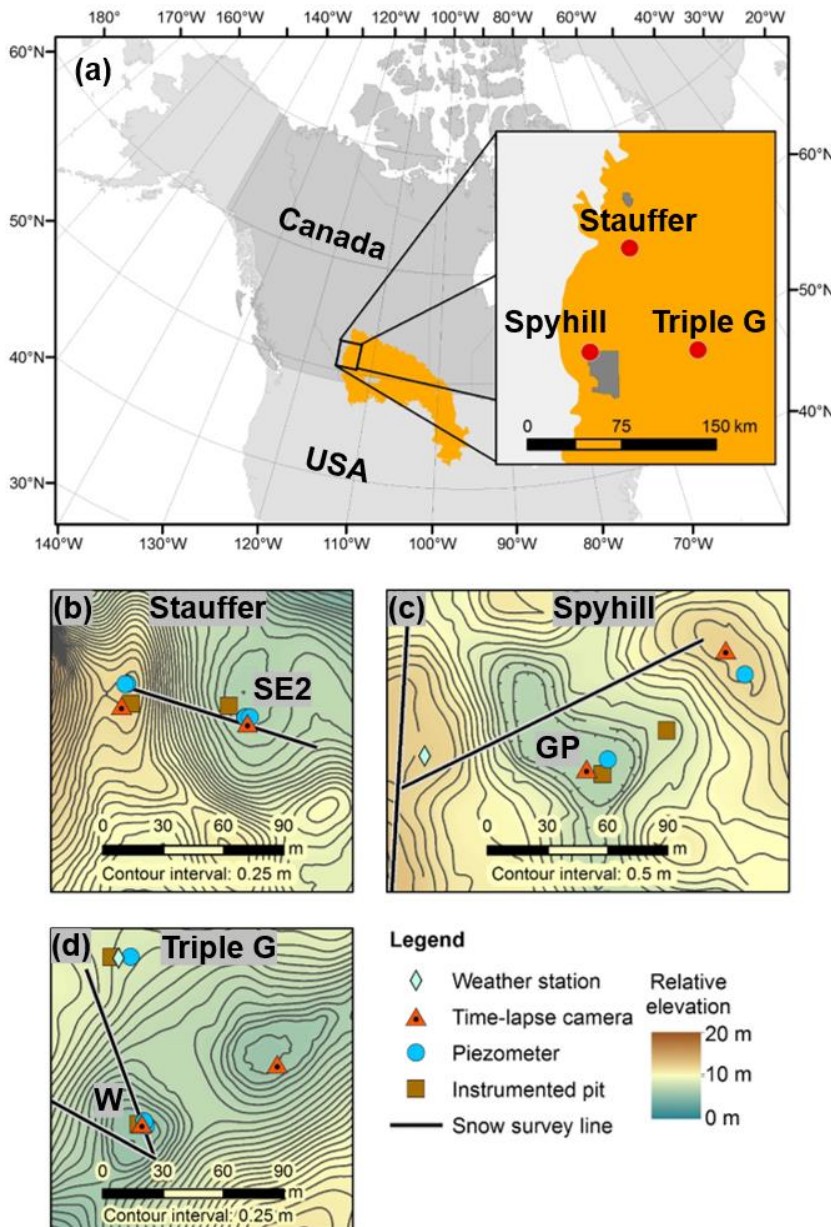

Figure 1: Location of (a) Northern Prairie region and study sites, and plan view of instrumentation and relative elevations of (b) Stauffer site, (c) Spyhill site, and (d) Triple G site.





**Figure 2: Temporal variability of hydrologic variables during the monitoring period for Stauffer upland location showing (a) average daily temperature and identification of snowmelt events; (b) soil temperatures; (c) liquid soil moisture, with pre-freezing values shown as circles; and (d) groundwater level measured in piezometers, screen depth indicates midpoint of screen interval. In panel (b), blue shading indicates frozen ground, purple indicates partially frozen, and red indicates unfrozen ground.**




**Figure 3: Temporal variability of hydrologic variables during the monitoring period for depression SE2 at Stauffer showing (a) average daily temperature and identification of snowmelt events; (b) depression pond level, gray circles indicate manual reading from snow gauges and black line is pond depth measured by a pressure transducer, recession rate during Spring also shown (dashed line); (c) soil temperatures; (d) liquid soil moisture, with pre-freezing values shown as circles; and (e) groundwater level measured in piezometers, screen depth indicates midpoint of screen interval. In panel (b), blue shading indicates frozen ground, purple indicates partially frozen, and red indicates unfrozen ground.**




**Figure 4: Temporal variability of hydrologic variables during the monitoring period for depression W at Triple G showing (a) average daily temperature and identification of snowmelt events; (b) depression pond level measured on snow gauges using time-lapse cameras, recession rate during Spring also shown (dashed line); (c) soil temperatures; (d) liquid soil moisture, with pre-freezing values shown as circles; and (e) groundwater level measured in piezometers, screen depth indicates midpoint of screen interval. In panel (b), blue shading indicates frozen ground, purple indicates partially frozen, and red indicates unfrozen ground.**





**Figure 5: Dates of snowcover depletion and timing of pond level increases in depression W at Triple G. The red circles indicate date most snowcover depletion occurred by. Gray lines show average hourly air temperature and blue lines show pond level.**



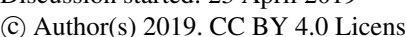

**Figure 6: Temporal variability of hydrologic variables during the monitoring period for Triple G upland location showing (a) average daily temperature and identification of snowmelt events; (b) soil temperatures; (c) liquid soil moisture, with pre-freezing values shown as circles; and (d) groundwater level measured in piezometers, screen depth indicates midpoint of screen interval. In panel (b), blue shading indicates frozen ground, purple indicates partially frozen, and red indicates unfrozen ground.**





**Figure 7: Temporal variability of hydrologic variables during the monitoring period for Spyhill GP depression showing (a) average daily temperature and identification of snowmelt events; (b) depression pond level measured by a pressure transducer, recession rate during Spring also shown (dashed line); (c) soil temperatures; (d) liquid soil moisture, with pre-freezing values shown as circles; and (e) groundwater level measured in piezometers, screen depth indicates midpoint of screen interval. In panel (b), blue shading indicates frozen ground, purple indicates partially frozen, and red indicates unfrozen ground.**




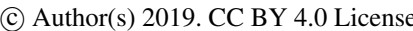

**Figure 8: Temporal variability of hydrologic variables during the monitoring period for Spyhill upland location showing (a) average daily temperature and identification of snowmelt events; (b) soil temperatures; (c) liquid soil moisture, with pre-freezing values shown as circles; and (d) groundwater level measured in piezometers, screen depth indicates midpoint of screen interval. In 5 panel (b), blue shading indicates frozen ground, purple indicates partially frozen, and red indicates unfrozen ground.**





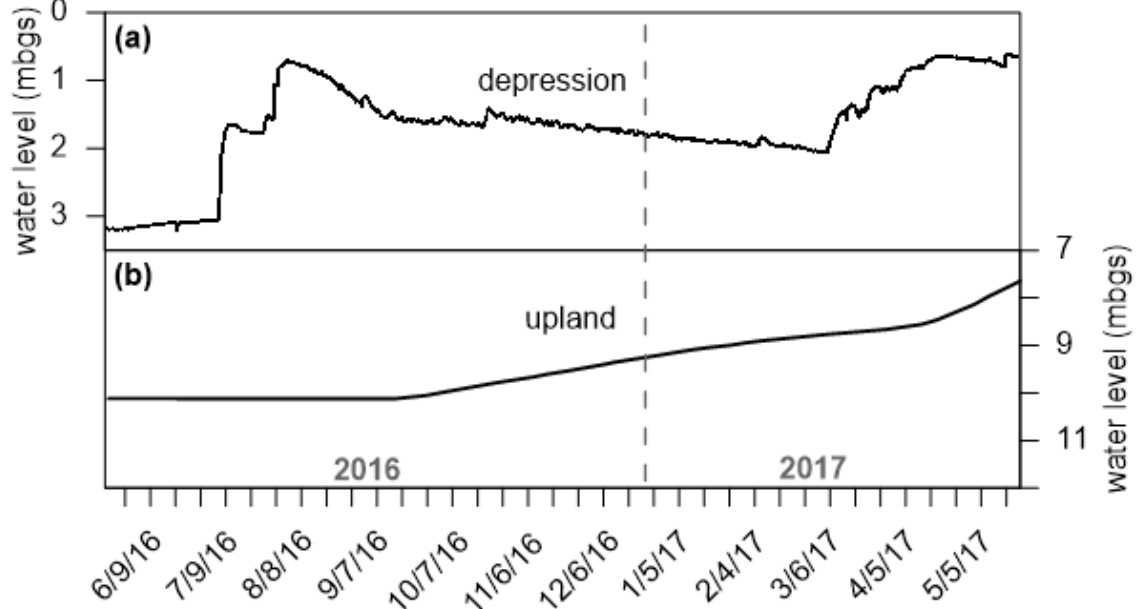

**Figure 9: Groundwater levels in piezometers at Spyhill (a) depression GP and (b) upland from June 2016 to mid-May 2017. Gray dashed line indicates the beginning of 2017.**