# Peer review of "Effects of preferential flow on snowmelt partitioning and groundwater recharge in frozen soils"

_Hydrology and Earth System Sciences, 2019_

## Referee Comment (RC1) · Andrew Ireson (Referee) · 26 Jun 2019

This is a field study of infiltration in frozen soils subject to mid-winter melts. This is a highly complex process with pathways that switch on and off at different times and places, and where at times there appears to be siginficant bypass flow occuring. This field study is extremely well designed to capture the detailed aspects of these processes. This study relies on qualitative insights from field observations, particularly looking at the location and timing of responses to melt in the soils, groundwater and ponds. Very nice observations of water content rises below the frost zone are presented. I think the understanding that is laid out in this paper is consistent with a range

of previously reported insights - such as the surprising absence of runoff from snowmelt and the isotopic signatures of groundwater. The paper presents an outstanding synthesis of knowledge in this area and should be required reading for anyone trying to model this environment. The paper is generally very well written. I think it could have been about 1/3 shorter. The results section in quite a labor to read through, but I don't have recommendations to improve this - I think it is necessary to explain in extensive detail how the different instruments respond during the melt period. The introduction is well written and covers the relevant literature well. The study objective is clear. Experimental methods and instrumentation are good. Methods are good. The discussion is again slightly long winded in style, but excellent in it's coverage of the insights and literature. I think there is a solid contribution here, and the paper should be published. I note below some minor comments. The main improvement would be to more clearly explain where the soil pits are located in the depressions, and including cross-sections would help here alot.

P. 1, L. 15: "the role of shallow subsurface flow" - I'm being slightly pedantic here, but this is ambiguous language... say explicitly what "the role" is - e.g. that there is a shallow lateral subsurface transmission pathway through the frozen soil, from uplands to depressions... if that's what is meant? Maybe appropriate to use the term "interflow".

P. 1, L. 17: "before ground thaw" - do you mean total thaw, or the commencement of thaw? This could be stated more clearly and precisely.

P. 2, L. 22: It could be appropriate here to note that zero-till cropping, which is in widespread useage for the past maybe 20 years in the prairies, might also allow macropores to be preserved.

P. 4, L. 2: confusing - should "infiltrates within" be "runs off into and then infiltrates beneath"? I think the point that the pond water level rises are not corrected for the volume of infiltration below the pond during the period of runoff. It would seem reasonable to ignore this likely small error (as the authors have done).

P. 4, L. 27: It's important to say whether or not a the soil pit was installed below the depression or adjacent to the depression. This is critical to the interpretation of pond water recessions juxtaposed against the soil temperatures. This is unclear to me from the text and from Figure 1. Cross-sections in Figure 1 would be extremely helpful to interpret where the measurements are taken from, including piezometer depths.

P 6., L. 31: Your data in Figure 2 and 3 show the water content responding to the Spring melt event before the temperature responds. Why?

P. 6, L. 32: at Stauffer the increase in RR between MW3 and spring (31 and 33) seems negligible and well within likely error bounds - this point should be acknowledged. The increases are far more convincing at Triple G and Spyhill and maybe there is a reason for that?

P. 7, L. 26: This paragraph describes the data in Figure 4, but studying Figure 4 it does not appear correct in a number of places - specifically the first and third sentences.

―――――――――――――――――――

---

## Referee Comment (RC2) · Anonymous Referee #2 · 24 Jul 2019

The manuscript of Mohammed at al. describes a field study of winter time infiltration in a cold region environment in the Canadian prairies where spring snowmelt and mid-winter melt events can have an important contribution to groundwater recharge. They used a sensor setup at three sites, each with two installed soil pits, including measurements of soil moisture, soil temperature and groundwater head. Furthermore, snow surveys were conducted and ponding levels in the depressions were recorded. The authors observed high amounts of infiltration during times when the soil was still frozen and corresponding increases in soil moisture content, some below the frost layer or non-sequential sensor reactions. Additionally, the recession rate of the water in the depression indicated infiltration into the frozen soils. The authors concluded that preferential flow in previous air-filled macropores are the dominant mechanism that allows water under frozen soil conditions to infiltrate. Increased runoff ratios with multiple midwinter melt events indicate refreezing of the infiltrating water and increased surface runoff. A time delay of snowmelt and ponding of water in the depressions was interpreted as lateral shallow subsurface flow into the depressions.

The study faces a highly relevant topic since many studies showed the reduction in hydraulic conductivities of the soil matrix under frozen conditions (e.g. Kane & Stein 1983, doi:10.1029/WR019i006p01547), but others observed that frozen soil still allows water to infiltrate (e.g. van der Kamp, doi:10.1002/hyp.1157). Frozen soil infiltration is a complex topic which includes the heterogeneity of soil flow processes as well as the effect of the soil thermal state and phase change of water. The role of preferential flow in frozen soil infiltration had become more attention in the last years (e.g. Holten et al. 2019, doi:10.2136/vzj2018.11.0201; Demand et al. 2019, doi:10.2136/vzj2018.08.0147; Watanabe & Kugisaki 2016, doi:10.1002/hyp.10939), studies which clearly show that frozen soil should not be treated as impermeable layers. How preferential flow in frozen soils can influences the magnitudes of water balance components over a winter season with multiple snowmelt events on a larger scale (>plot scale) is unclear till now. Therefore, the study of Mohammed et al. gives interesting insights and should be published, but some revisions are needed.

The study fits into the scope of the journal, is well written with a good introduction and clear objectives. The manuscript cites the most relevant literature and has appropriate methods for studying the phenomenon of frozen soil infiltration. I agree with the comments of Referee 1 (Andrew Ireson) and have some additional general comments mainly regarding the length, structure and argumentation of the discussion.

I suggest to restructure and shorten section 5.1 and 5.2 of the discussion. It is rather long for the main findings of the study and repeats many points. For example P10 L1-26 discusses preferential flow for the argumentation of snowmelt partitioning. However, preferential flow is again discussed in a similar way on P11 L20 - P12 L19. I suggest

to summarize these parts. Furthermore, I suggest to move the Table 3 and arguments on P11 L20-24 to the results (as it also reads like results). Table 3 is not mentioned in the results and the authors already discuss the relevance of preferential flow without pointing out its evidence in the results. Additionally, the authors should give some more information on the soil properties (if possible). Especially porosity seems to be important in the context of the study, because air-filled porosity is used for the argumentation of preferential flow. E.g. antecedent soil moisture in relation to porosity can be used as an estimate of air-filled porosity available for infiltration.

Please make sure that you be consistent with the description in the results. For example you do not mention MW2 for the Spyhill Upland. Furthermore, sometimes it is not completely clear if you still talk about the upland or the depression (e.g. P6 L27-29).

Specific comments:

P3 L16: Change ". . .of the region surrounding the study sites . . ." to ". . . of the study sites. . ."

P4 L13: Please specify why it was not always possible to use pressure transducers.

P5 L10-11: How many soil cores were taken and at which depth? I think you first mention this in the results. Why has the Ksat using a permeameter been only determined for this site? Specify that all Ksat measures were performed for unfrozen soil.

P5 L17-18: The slug tests were done for all sites? Please specify this.

P5 L24: Please clarify that even if Ro is underestimated (P4 L23-24) the catchment wide infiltration rate I is correct. An underestimated Ro leads to an overestimation of I at the uplands, but an underestimation of I in the depressions. From the pure equation one can think that the calculated I also contains the error resulting from the R0 estimation (hence I would be too high).

P6 L15 and Table 2: Be more consistent for Stauffer MW1 and use "-" for all components or leave the event out since it was not calculated.

P7 L5-6: Again, I think how you calculated Ksat and that the permeameter tend to underestimate Ksat (with a reference) should be mentioned in the methods.

P7 L22 and Table 2: Why did Triple G had no change of SWE during the spring melt event?

P8 L8-11: Be careful with the comparison of Ksat and recession rates, since a unit gradient assumption is questionable at Triple G with 50 cm ponding head.

P9 L23-28: The runoff ratios also depend on the snowmelt rates observed during a certain event or year. For a comparison of different sites, events or years it is important to mention this.

P10 L16: Delete "subsurface" in "subsurface infiltration/refreezing"

P10 L28: How can you be sure that these lateral pathways are preferential pathways?

P11 L28: Maybe Graham & Lin (2011) (doi: 10.2136/vzj2010.0119) is a better reference.

P12 L10-11: Why do you know that infiltration was limited by the hydraulic conductivity of the zone beneath the frozen layer and not by the frozen layer itself? You do not know the saturated hydraulic conductivity of the frozen layer and from the water level recessions you can just estimate an integrated Ksat (with unknown extend) by assuming a unit gradient (what's not always the case).

P12 L15-16 I would add a reference here, e.g. Schwen et al. (2011) (doi:10.1016/j.still.2011.02.005)

P12 L30-31 I would argue that the infiltration is rather dependent on the amount of connected and air-filled macropores in the frozen layer than on the infiltration rate of the subsoil. Without connected macropores the infiltration rate of the topsoil would be much lower since only the frozen soil matrix would conduct water and hence it would take the water a long time to even reach the subsoil.

P13 L31: Change to "... during midwinter snowmelt enhanced surface runoff generation..."

P14 L23: Data availability: Delete "and". Furthermore, there is no Table in the Supplements.

---

## Author Comment (AC1) · 2 Sep 2019

We thank Dr. Andrew Ireson for his positive review and constructive suggestions, which allowed us to improve the clarity of the manuscript. In the following section, reviewer comments are in bold, author responses in regular font, and changes made to the text are in italics.

Authors response to specific comments:

**P. 1, L. 15: "the role of shallow subsurface flow" - I'm being slightly pedantic here, but this is ambiguous language... say explicitly what "the role" is - e.g. that there is a shallow lateral subsurface transmission pathway through the frozen soil, from uplands to depressions... if that's what is meant? Maybe appropriate to use the term "interflow".**

We acknowledge this statement was a bit ambiguous. We agree that a more explicit statement is needed and have included the suggested wording of 'a shallow lateral subsurface transmission pathway through the frozen soil' and also included the term 'interflow'. The sentence has been modified to:

*At one site, time lags of up to 3 days between snowcover depletion on uplands and ponding in depressions demonstrated the role of a shallow subsurface pathway or interflow through frozen soil in routing snowmelt from uplands to depressions.*

**P. 1, L. 17: "before ground thaw" - do you mean total thaw, or the commencement of thaw? This could be stated more clearly and precisely.**

We agree and have rephrased to 'complete ground thaw' to be more precise.

**P. 2, L. 22: It could be appropriate here to note that zero-till cropping, which is in widespread usage for the past maybe 20 years in the prairies, might also allow macropores to be preserved.**

We agree and have added a sentence on this.

'*... have a higher frozen soil infiltrability due to the presence and development of a macropore network, in comparison to tilled croplands, where annual cultivation breaks up the macropore network near surface (van der Kamp et al., 2003). In addition, zero-tillage cropping, which has been increasingly adopted in the Prairies over the past several decades, may also allow macropore networks to be preserved (Tiessen et al., 2010).*'

Additional reference: Tiessen, K.H.D., Elliott, J.A., Yarotski, J., Lobb, D.A., Flaten, D.N. and Glozier, N.E. 2010. Conventional and conservation tillage: Influence on seasonal runoff, sediment, and nutrient losses in the Canadian prairies. *Journal of Environmental Quality* **39**:964-980.

**P. 4, L. 2: confusing - should "infiltrates within" be "runs off into and then infiltrates beneath"? I think the point that the pond water level rises are not corrected for the volume of infiltration below the pond during the period of runoff. It would seem reasonable to ignore this likely small error (as the authors have done).**

We have made the change as per Dr. Ireson's suggestion. We have also stated more explicitly that the pond level rises are not corrected for the volume lost to infiltration below the pond. The sentence has been modified to:

'*This method underestimates runoff as it does not consider the volume of water that initially runs off into and then infiltrates beneath the depression prior to observable ponding (Hayashi et al., 2003).*'

**P. 4, L. 27: It's important to say whether or not the soil pit was installed below the depression or adjacent to the depression. This is critical to the interpretation of pond water recessions juxtaposed against the soil temperatures. This is unclear to me from the text and from Figure 1. Cross-sections in Figure 1 would be extremely helpful to interpret where the measurements are taken from, including piezometer depths.**

We agree this is an important point. We have added text to clarify that the soil pit was installed directly below the lowest surface elevation point in the depression. We have also added a cross-section as requested. To address this concern and those of Reviewer 2 (regarding providing more subsurface information), we have split Figure 1 into two separate figures with additional information. The new Figure 1 has the location of the study region, and soil and sediment cover for the region. The new Figure 2 provides the plan view map of the sites along with a cross-section showing subsurface instrumentation. We have also included a new figure with depth-$K_{sat}$ profiles for the three sites.

**P 6., L. 31: Your data in Figure 2 and 3 show the water content responding to the Spring melt event before the temperature responds. Why?**

We thank Dr. Ireson for mentioning this issue. During the Spring melt event, soil water contents increase before the soil temperatures respond (at the same depths) due to the 'zero-degree' curtain effect, in which latent heat transfer prevents soil temperatures from rising above 0 °C until pore-ice has completely melted. As such, a water content response while temperature at the same depth remains at 0 °C indicates porewater phase change is occurring. In other words, soil frost is present and likely thawing. We have added discussion of this point to the text.

**P. 6, L. 32: at Stauffer the increase in RR between MW3 and spring (31 and 33) seems negligible and well within likely error bounds - this point should be acknowledged. The increases are far more convincing at Triple G and Spyhill and maybe there is a reason for that?**

We agree and acknowledge that the increase in RR at Stauffer between MW3 and Spring is considered negligible compared to Triple G and Spyhill and have made a statement acknowledging this. We are not entirely certain what the cause of this discrepancy is, but we can speculate that one possible reason is that soils at SE2 have more sand content than the other two sites which are more clay-rich. Thus, in addition to macropore flow, enhanced matrix flow and/or imbibition from macropores to the matrix in frozen soil at SE2 maybe allow more faster

drainage to the subsoil during snowmelt and keep the near surface soil relatively permeable prior the next melt event. Another possible explanation is that SE2 may have been subject to some overflow (i.e. fill and spill) to the north of the depression. The relatively stable RR values for SE2 over MW2 and Spring could also be a combination of both factors (sandy soil and some overflow potential).

**P. 7, L. 26: This paragraph describes the data in Figure 4, but studying Figure 4 it does not appear correct in a number of places - specifically the first and third sentences.**

We thank Dr. Ireson for bring up these discrepancies. We assume that the mistake in the first sentence is that soil moisture increased slightly at all depths 0.2 to 0.8 m, but that only the 0.2 m sensor rose above its pre-freezing value. The sentence has been modified to correct this:

*Liquid soil moisture at depths 0.2-0.8 m under the depression gradually increased between events MW1 and MW2 (Fig. 4d) along with a slight response in groundwater level in the piezometer, which was dry prior to MW1 (Fig. 4e), indicating some infiltration had taken place.*

We assume that the discrepancy in the third sentence is that the soil moisture sensors at 0.2 m does actually show a gradual response after MW3, suggesting that they're might be some infiltration taking place to that depth. The sentence has been modified to correct this:

*After MW3, liquid soil moisture at 0.2 m increased gradually, but no event-specific changes in liquid soil moisture and temperature occurred below that depth (Fig. 4d) despite the rise in the pond level (Fig. 4b). This suggests little infiltration in the depression occurred during this event due to prior refreezing of meltwater in the soil profile following MW2.*

---

## Author Comment (AC2) · 2 Sep 2019

We thank the Reviewer for their positive review and constructive suggestions, which allowed us to improve the clarity of the manuscript. In the following response, reviewer comments are in bold font, author responses in a regular font, and changes made to the text are in italic font.

Authors response to general comments:

**I suggest to restructure and shorten section 5.1 and 5.2 of the discussion. It is rather long for the main findings of the study and repeats many points. For example P10 L1-26 discusses preferential flow for the argumentation of snowmelt partitioning. However, preferential flow is again discussed in a similar way on P11 L20 - P12 L19. I suggest to summarize these parts.**

We agree with the reviewer that section 5.1 and 5.2 can be written more concisely, Dr. Ireson has also made this point. We have made an effort to reduce the length of these sections and remove redundant points.

**Furthermore, I suggest to move the Table 3 and arguments on P11 L20-24 to the results (as it also reads like results). Table 3 is not mentioned in the results and the authors already discuss the relevance of preferential flow without pointing out its evidence in the results.**

We thank the reviewer for this point, but we believe the that best place for Table 3 is in its current section of 5.2 which specifically discusses depression and ponded infiltration dynamics. The purpose of Table 3 is to summarise the collective evidence for preferential flow occurring during depression-focused infiltration and groundwater recharge at the study sites, without having to re-discuss points in the results section. Furthermore, we don't discuss why these observations are evidence for preferential flow until section 5.2, so we believe it would be out of context in the results section.

**Additionally, the authors should give some more information on the soil properties (if possible). Especially porosity seems to be important in the context of the study, because air-filled porosity is used for the argumentation of preferential flow. E.g. antecedent soil moisture in relation to porosity can be used as an estimate of air-filled porosity available for infiltration**.

We agree with the reviewer that information on porosity is important and we have included an additional table including soil information (porosity, bulk density, grain size distribution, $K_{sat}$). Additionally, to address this comment and those of Dr. Ireson, we have split Figure 1 into two separate figures with additional soil and sediment information. We have also included a new figure with depth-$K_{sat}$ profiles for the 3 sites.

**Please make sure that you be consistent with the description in the results. For example, you do not mention MW2 for the Spyhill Upland. Furthermore, sometimes it is not completely clear if you still talk about the upland or the depression (e.g. P6 L27-29)**

We thank the reviewer for catching this oversight. We have added some discussion of MW2 to the Spyhill results section 4.3 stating that incomplete snowcover depletion at GP during MW2 prevented the closure of the water balance, but no runoff was observed. We have also gone through all results sections to ensure it is clear in each section which landscape position we are referring to.

Author response to specific comments:

**P3 L16: Change ": : :of the region surrounding the study sites : : :" to ": : : of the study sites: : :"**

Changed.

**P4 L13: Please specify why it was not always possible to use pressure transducers.**

Transducers were not able to be deployed during MW events due to the possibility of them being damaged by freezing of water in the pond. We have added text to the methods section explaining this.

**P5 L10-11: How many soil cores were taken and at which depth? I think you first mention this in the results. Why has the Ksat using a permeameter been only determined for this site? Specify that all Ksat measures were performed for unfrozen soil.**

$K_{sat}$ at depths below the frost zone (2-3 m) were only taken at SE2 because of the water table depth at this site. A piezometer at 3m below the ground surface was installed at this site but the water table is generally more than 5 m below ground surface in the depression and the piezometer remained dry except for a few days during snowmelt. Thus, no slug test was able to be performed at the depth right below the frost zone (i.e., 3 m piezometer) and lab permeameter results were used to fill this gap in the data. We have added text stating this.

We have also specified that all $K_{sat}$ measurements were performed under unfrozen conditions.

**P5 L17-18: The slug tests were done for all sites? Please specify this.**

At the time of the initial submission, slug tests were not carried out in the shallowest piezometers in depressions SE2 (Stauffer) and W (Triple G) due to the fact that those piezometers tended to be dry (other than during snowmelt) or had water levels within the screened interval of the piezometer, which would make slug test results unreliable. Since submission of the original manuscript, the piezometer in depression W was slug tested when conditions were more favorable, and this value ($3\times10^{-7}$ m s$^{-1}$) has been included in the updated text and subsequent discussion. See previous comment regarding SE2.

**P5 L24: Please clarify that even if Ro is underestimated (P4 L23-24) the catchment wide infiltration rate I is correct. An underestimated Ro leads to an overestimation of I at the uplands, but an underestimation of I in the depressions. From the pure equation one can**

**think that the calculated I also contains the error resulting from the R0 estimation (hence I would be too high).**

We thank the reviewer for bringing up this inciteful point. We have added text discussing this point.

'*In the formulation of equation 1, $R_o$ is slightly underestimated, which leads to an overestimation of I under uplands, but also underestimates I in the depression, thus we believe this is a reasonable estimate of the catchment wide I.*'

**P6 L15 and Table 2: Be more consistent for Stauffer MW1 and use "-" for all components or leave the event out since it was not calculated.**

We have included measured values of precipitation and vapour flux measured over the period and left '-' for the components which could not be measured or estimated.

**P7 L5-6: Again, I think how you calculated Ksat and that the permeameter tend to underestimate Ksat (with a reference) should be mentioned in the methods.**

We have included a statement and reference on the tendency of small core-permeameters to underestimate field $K_{sat}$.

Reference: Schulze-Makuch, D., Carlson, D. A., Cherkauer, D. S., and Malik, P. 1999. Scale dependency of hydraulic conductivity in heterogeneous media. *Groundwater*, **37**(6): 904-919.

**P7 L22 and Table 2: Why did Triple G had no change of SWE during the spring melt event?**

A *ΔSWE* value of zero means that at the start of the time period over which the water balance was calculated before the Spring event, there was no snowcover on the ground surface at the site. During MW2 all snowcover was depleted and so the measured precipitation that fell from the beginning time of the calculation to the end of the snowmelt period would be the total input.

**P8 L8-11: Be careful with the comparison of Ksat and recession rates, since a unit gradient assumption is questionable at Triple G with 50 cm ponding head.**

We acknowledge that a unit gradient assumption may be questionable under ponding conditions, however the reason we make this comparison is only to note that the rate is the same order of magnitude as measured $K_{sat}$ values. While ponding conditions may create significant hydrostatic pressure at the centre of the pond, gradients at the edge of the pond are likely significantly less than unity. Moreover, it is unlikely that ponding head in saturated partially-frozen soil would create a hydraulic gradient large enough (~10) to increase the recession rate to an order of magnitude above $K_{sat}$. For example, we can compare infiltration in the ponded depression to a much-simplified analog of a single-ring infiltrometer. The increased infiltration flux due to ponding would be roughly ponded head, H, divided by [0.6×pond radius]. Thus, a 10 m diameter pond with 0.50 m pond height, would increase flux by 0.5/(0.6*5) = 0.17 i.e. 17% or a

factor of 1.17. We acknowledge this is not a perfect comparison, but it shows the effect is relatively small.

**P9 L23-28: The runoff ratios also depend on the snowmelt rates observed during a certain event or year. For a comparison of different sites, events or years it is important to mention this.**

We thank the reviewer for bringing this point up and agree that it should be mentioned when comparing runoff ratios. We have added a sentence to the discussion bringing up this point.

'*In addition to frozen soil infiltrability, runoff ratios at any given site also depend on snowmelt rates during a melt event, however some useful insight can be made from comparing runoff ratios here with other studies. The snowmelt runoff ratios from all sites ranged between 0 and 57%, with a mean of 21%. These values are lower than those reported by Hayashi et al. (1998) for a…*'

**P10 L16: Delete "subsurface" in "subsurface infiltration/refreezing"**

Change made.

**P10 L28: How can you be sure that these lateral pathways are preferential pathways?**

This is a very good point, and the reviewer is correct in stating we cannot be certain that the lateral pathways are preferential. However, there were a few factors that we considered when making this argument, which were not included in the manuscript for the sake of brevity. Firstly, the soil under the upland areas at Triple G were relatively dry and never reached close to saturation during overwinter and spring snowmelt periods. Saturated or nearly saturated conditions would be needed to produce diffuse (non-preferential flow) lateral subsurface flow along a topographic gradient in the near surface soils. However our argument is that if most of the infiltrated water flows through previously air-filled macropores surrounded by a matrix of reduced permeability due to freezing conditions, then macropores would require less water to fill (compared to total porosity), and an impeding layer at some shallow depth in the frozen soil (higher soil frost and/or decreased macroporosity) may cause some of the infiltrated water to be laterally deflected along the topographic gradient to depressions.

Secondly, we also performed a simple calculation assuming diffuse saturated flow:

Estimation of volume pond increased by over Spring melt event:

   Height pond increased by = 0.1 [m] (Spring event)
   Volume of water added to pond = ~74 [$m^3$] (volume calculated from pond height using depth-volume relationship from topographic survey of depression W)

Volumetric flux = $K_{sat}$ × saturated thickness × average slope of catchment × pond circumference

   $K_{sat} = 1.0×10^{-3}$ [m s$^{-1}$] assumed
   saturated thickness = 0.1 [m] assumed

Slope = 0.08 [-] estimated from topographic surveys
Diameter of wetted area = ~30 [m] from satellite imagery on March 20, 2017
Circumference of pond = 94.25 [m]

Volumetric flux = ~65 [$m^3 d^{-1}$]

This calculation is very rough and makes several simplifying assumptions but it shows that for saturated diffuse subsurface flow to transport the observed volume of water over the time delays we observed (1 to 2 days), the hydraulic conductivity of the near surface would have to be one to two orders of magnitude higher ($10^{-3}$ m $s^{-1}$) than the near surface hydraulic conductivity measured with single ring infiltrometers (circa. $10^{-5}$-$10^{-6}$m $s^{-1}$). These near surface $K_{sat}$ values are included in the new table and figure mentioned above.

Furthermore, we acknowledge that identifying mechanisms of flow governing subsurface runoff or interflow is very complicated as direct observation of the flowpath is almost impossible. We are currently carrying out chemical tracer experiments at Triple G to better understand this subsurface transmission pathway.

**P11 L28: Maybe Graham & Lin (2011) (doi: 10.2136/vzj2010.0119) is a better reference**.

Thanks, we have added the reference to text and reference list.

**P12 L10-11: Why do you know that infiltration was limited by the hydraulic conductivity of the zone beneath the frozen layer and not by the frozen layer itself? You do not know the saturated hydraulic conductivity of the frozen layer and from the water level recessions you can just estimate an integrated Ksat (with unknown extend) by assuming a unit gradient (what's not always the case).**

The reviewer is right that we do not know that hydraulic conductivity of the frozen layer, especially as it changes with time due to changes in total water and ice content at different depths. Our reasoning for stating that the infiltration rate was limited by the hydraulic conductivity of the zone beneath the frozen layer was that the recession/infiltration rate was the same order of magnitude as the $K_{sat}$ measured beneath the frost zone (either by slug tests or permeameters), but was consistently an order of magnitude lower than the $K_{sat}$ of the near surface soil (0-0.3 m) measured by single ring infiltrometers or Guelph permeameters at the sites. Given that ponding conditions saturate the entire soil profile (temporarily), infiltration would be limited by the lowest hydraulic conductivity layer, which in this case was the zone below the frost zone. However, we acknowledge that if macropores are blocked with ice, then the frost layer hydraulic conductivity would be the limiting factor. We have modified the text to make this point more clearly.

**P12 L15-16 I would add a reference here, e.g. Schwen et al. (2011) (doi:10.1016/j.still.2011.02.005)**

We thank the reviewer for this reference and have added it to the text and references.

**P12 L30-31 I would argue that the infiltration is rather dependent on the amount of connected and air-filled macropores in the frozen layer than on the infiltration rate of the subsoil. Without connected macropores the infiltration rate of the topsoil would be much lower since only the frozen soil matrix would conduct water and hence it would take the water a long time to even reach the subsoil.**

We completely agree with the reviewer. Our argument that the infiltration rate was dependent on the $K_{sat}$ of the subsoil was precisely because it was not being limited by the frozen layer above because of the presence of conductive macropores. However, as mentioned above, we acknowledge that if macropores are blocked with ice, then the frost layer hydraulic conductivity would be the limiting factor and have modified the text accordingly.

**P13 L31: Change to ": : during midwinter snowmelt enhanced surface runoff generation: : :"**

Change made.

**P14 L23: Data availability: Delete "and". Furthermore, there is no Table in the Supplements.**

We apologise for this oversight and will include the table in the updated supplement.